



# Convective suppression before and during the United States Northern Great Plains Flash Drought of 2017

Tobias Gerken[1,2], Gabriel T. Bromley[1], Benjamin L. Ruddell[2], Skylar Williams[1], and Paul C. Stoy[1]

[1]Montana State University, Department of Land Resources and Environmental Sciences, Bozeman, MT 59717, USA
[2]Northern Arizona University, School of Informatics, Computing, and Cyber Systems, Flagstaff, AZ 86011, USA

*Correspondence to:* Tobias Gerken (tobias.gerken@montana.edu)

**Abstract.** Flash droughts tend to be disproportionately destructive because they intensify rapidly and are difficult to prepare for. We demonstrate that the 2017 U.S. Northern Great Plains (NGP) flash drought was preceded by a breakdown of land-atmosphere coupling. Severe drought conditions in the NGP were first identified by drought monitors in late May 2017 and rapidly progressed to exceptional drought in July. The likelihood of convective precipitation in May 2017 in northeastern Montana, however, resembled that of a typical August when rain is unlikely. Based on the lower tropospheric humidity index ($HI_{low}$), convective rain was suppressed by the atmosphere on nearly 50% of days during March in NE Montana and central North Dakota, compared to 30% during a normal year. Micrometeorological variables, including potential evapotranspiration, were neither anomalously high nor low before the onset of drought. Incorporating convective likelihood to drought forecasts would have noted that convective precipitation in the NGP was anomalously unlikely during the early growing season of 2017. It may therefore be useful to do so in regions that rely on convective precipitation.

## 1 Introduction

Rapid onset "flash" droughts (Otkin et al., 2017) due to extreme heat or precipitation deficit (Mo and Lettenmaier, 2016) are difficult to predict and prepare for, and thus tend to be disproportionally destructive (Ford and Labosier, 2017). An unprecedented flash drought took place across parts of the U.S. Northern Great Plains (NGP) and Canadian Prairie Provinces during the 2017 vegetative growing season (Fig. 1) that in some areas was the worst in recorded history.

Even though abnormally dry conditions were reported by agricultural producers in northeastern Montana as early as late April (Montana DNRC, 2017), the 2017 NGP flash drought was not foreseen in the seasonal forecast. The U.S. National Weather Service Climate Prediction Center's (CPC) three-month (JJA) seasonal forecast issued on May 18, 2017 reported above average precipitation probabilities across the NGP. Severe drought conditions (D2 as classified by the U.S. Drought Monitor, USDM, Svoboda et al., 2002) began in late May 2017 in the central Dakotas and then extended westward toward Montana (Fig. 1; Tab. 1). At the peak of the drought in early September, nearly 3/4 of Montana was under *extreme* (D3) or





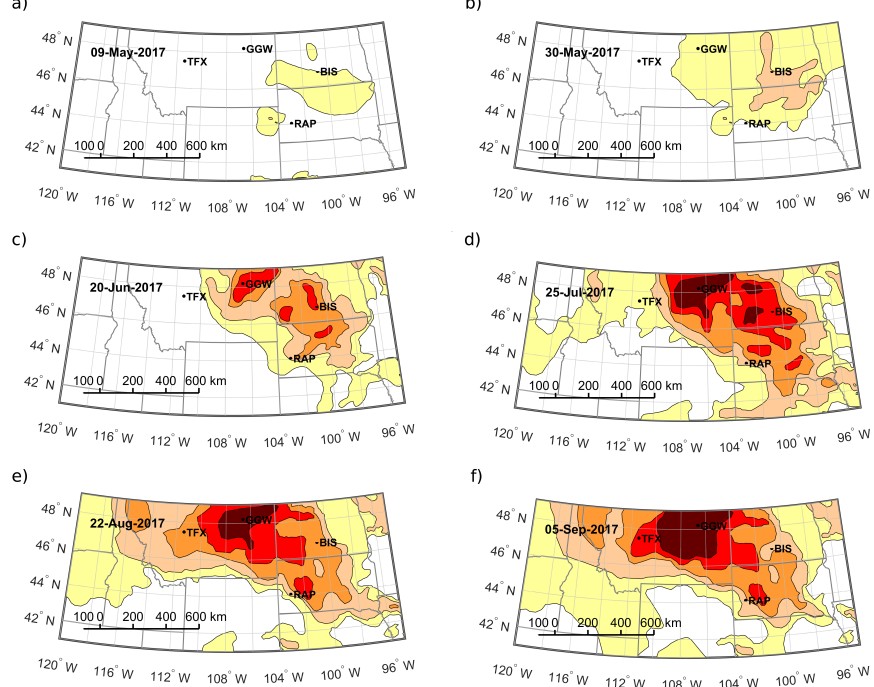

**Figure 1.** The U.S. Drought Monitor classification of the 2017 Northern Great Plains Flash Drought during the weeks of (a) 09-May-2017; (b) 30-May-2017; (c) 20-Jun-2017; (d) 25-Jul-2017; (e) 22-Aug-2017; and (f) 05-Sep-2017. The locations of atmospheric sounding stations at Bismarck, Rapid City, Glasgow and Great Falls are indicated. The dates were chosen according to drought development stages at the sites. D0 (*abnormally dry*) -– yellow; D1 (*moderate drought*) -– tan; D2 (*severe drought*) — orange; D3 (*extreme drought*) -– red; D4 (*exceptional drought*) -– dark red.

*exceptional* (D4) drought. Nearly 2/3 of North Dakota was under *severe* to *exceptional* drought in mid-August, and more than half of South Dakota was under *severe* to *exceptional* drought in mid-July, causing acute and ongoing economic and ecological consequences across the region.

The USDM is a diagnostic product that combines the Palmer Drought Severity Index (PDSI, Palmer, 1965; Alley, 1984)
5  with U.S. Geological Survey Stream Flow, soil moisture observations (Fan and van den Dool, 2004), the Standardized Precipitation Index (Guttman, 1998, 1999), and expert knowledge. As such and as intended, it provides a useful tool for tracking and displaying drought across the U.S. (Svoboda et al., 2002), but does not provide information about the meteorological conditions causing the drought. Here, we demonstrate that an established approach for diagnosing the likelihood of convective precipitation (Findell and Eltahir, 2003a, b) as applied to the NGP (Gerken et al., 2018) indicated that convective precipitation
10  was unlikely to occur in the regions impacted by drought as early as March 2017 due to an anomalously dry lower atmosphere. Locally triggered convective events are not the volumetrically dominant water source for precipitation – weather systems contribute approximately 60% of total warm season precipitation in the U.S. Great Plains (Carbone and Tuttle, 2008) – but provide





**Table 1.** Climate stations from Global Historic Climatology Network (GHCN) and onset of drought stages (D0-D4, according to U.S. Drought Monitor, USDM) for the 2017 drought.

| Location | Code[*] | GHCN ID | Coordinates | Onset D0 | Onset D1 | Onset D2 | Onset D3 | Onset D4 |
|---|---|---|---|---|---|---|---|---|
| Bismarck, ND | BIS | USW00024011 | N 46.783°; W 100.757° | 09-May | 30-May | 06-Jun | 20-Jun | — |
| Rapid City, SD | RAP | USW00024090 | N 44.043°; W 103.054° | 06-Jun[**] | 13-Jun[**] | — | — | — |
| Great Falls, MT | TFX | USW00024143 | N 47.793°; W 111.382° | 25-Jul | 08-Aug | 22-Aug | 05-Sep | — |
| Glasgow, MT | GGW | USW00094008 | N 48.241°; W 106.621° | 30-May | 06-Jun | 13-Jun | 20-Jun | 25-Jul |

[*] National Weather Service station code.

[**] The USDM reported *moderate drought* conditions (D1) for Rapid City during winter 2016-2017. On 02-May-2017 *abnormally dry* conditions (D0) no longer persisted at Rapid City.

a critical moisture source during the vegetative growing season. We argue that it may therefore be useful to include convective likelihood to drought monitors and forecasts in regions where convective precipitation is an important moisture source.

## 2 Data and Methods

Convective precipitation in the NGP is part of a *transitional* regime characterized by both wet and dry coupling (Findell and
Eltahir, 2003b). Conditions alternate between atmospheric control of convective precipitation during the late growing season and periods during which changes in surface energy balance partitioning can lead to crossings between mixed-layer height and the lifted condensation level during the early vegetative growing season (Gerken et al., 2018). Such crossings are considered a necessary but not sufficient condition for the development of convective precipitation (Juang et al., 2007a, b).

To diagnose convective likelihood, we use atmospheric profiles of air temperature ($T$) and humidity ($q$) following (Findell
and Eltahir, 2003a, b) to calculate the convective triggering potential (CTP) and the lower tropospheric humidity index ($\mathrm{HI_{low}}$), a measure of atmospheric dryness:

$$\mathrm{HI_{low}} = (T_{50} - T_{d,50}) + (T_{150} - T_{d,150}) \tag{1}$$

$\mathrm{HI_{low}}$ is based on Lytinska et al. (1976) and calculated from the sum of dewpoint depressions (the difference between $T$ and dewpoint temperature $T_d$) at 50 hPa and 150 hPa above the ground level. CTP is the energy released by a hypothetical saturated
air parcel that ascends from 100 hPa to 300 hPa above the ground level and is thus a measure of atmospheric instability (Findell and Eltahir, 2003a, b), which is closely related to convective available potential energy (CAPE).

Based on early morning sounding data in Illinois, USA, it was originally proposed that convection did not develop for $\mathrm{CTP} < 0\,\mathrm{J\,kg^{-1}}$ due to lack of available energy, whereas the lower atmosphere was too dry to develop boundary-later clouds and associated precipitation if $\mathrm{HI_{low}} > 15°C$ (Findell and Eltahir, 2003a, b). Later research established different thresholds for
convective likelihood across different regions (Ferguson and Wood, 2011; Roundy et al., 2012), and we have established that convective development is unlikely for $\mathrm{HI_{low}}$ greater than approximately 20°C at Glasgow, Montana (GGW) near the centroid





of the 2017 drought (Fig. 1) (Gerken et al., 2018). We use this value as indicative of the threshold beyond which convective precipitation is unlikely to develop for the study region.

We calculated CTP and $\mathrm{HI_{low}}$ for 12 UTC using daily vertical radiosonde profiles of $T$ and $q$ for the period 1987 to 2017 from the University of Wyoming Upper Air Sounding Archive (http://weather.uwyo.edu/upperair/sounding.html). Four stations

located in the NGP were selected: Bismarck, ND (Code: BIS, GHCN ID: 72764), Rapid City, SD (RAP, 72662), Great Falls, MT (TFX, 72776), and Glasgow, MT (GGW, 72768) (Table 1). Sounding operations at TFX commenced in September 1994 and data are not available beforehand.

Daily precipitation ($P$) and daily average $T$ data for the four locations (Tab. 1) were obtained from the Global Historic Climate Network (GHCN) Daily product described in Menne et al. (2012a, b). $T$ and $P$ data from Great Falls International

Airport were chosen for data instead of the adjacent Great Falls Weather Forecast Office (TFX, the sounding location) due to its longer data record that spans the analysis period. Daily data for the 30-year period from 1987 to 2016 were averaged to estimate the $T$ and $P$ climatology at the four study sites. $P$ occurring on February 29 was attributed to February 28 to compute the $P$ climatology.

Drought maps of the NGP and surrounding regions were created using geographic shape files downloaded from the US

Drought Monitor (Svoboda et al., 2002). The drought monitor uses a blend of data sources to classify drought status between *abnormally dry* (D0), as well as *moderate* (D1), *severe* (D2), *extreme* (D3), and *exceptional drought* (D4).

We also analyze potential evapotranspiration ($\mathrm{ET}_p$) given recent findings that previous flash droughts were preceded by anomalously high evaporative stress (Otkin et al., 2016; Ford and Labosier, 2017). In the absence of direct measurements, daily $\mathrm{ET}_p$ was estimated from standard weather data, obtained from the Integrated Surface Hourly Dataset, following the protocol

established by the Food and Agriculture Organization (FAO) and outlined in Allen et al. (1998). The FAO Penman-Monteith equation for a reference crop is:

$$\mathrm{ET}_p = \frac{0.408\,\Delta\,(R_n - G) + \gamma\frac{900}{T_2 + 273}\,u_2\,\mathrm{VPD}}{\Delta + \gamma\,(1 + 0.34\,u_2)}, \tag{2}$$

where $R_n$ is net radiation at the land surface, $G$ is soil heat flux and assumed to be zero at the daily timescale, $T_2$ and $u_2$ are the mean daily air temperature and wind speed at 2 m height, respectively, VPD is the vapor pressure deficit, $\Delta$ is the slope

of saturation vapor pressure curve, and $\gamma$ is the psychrometric constant. Missing climatic data was treated according to the protocol outlined in Allen et al. (1998), such that for example $R_n$ was derived using daily temperature range ($T_{max} - T_{min}$), estimated extraterrestrial radiation $R_a$, and an estimate for net longwave radiation based on the Stefan-Boltzmann law (See eqs. (20–26) & eqs. (35–40) in the FAO document (Allen et al., 1998) for details).

## 3 Results

A comparison between 30-year average $P$ totals and cumulative $P$ during 2017 reiterates that 2017 was much drier than average at all four sounding locations in the NGP (Fig. 2). In a typical year, more than 50% of the annual $P$ in the NGP occurs from April to June during the main vegetative growing season, but there is considerable interannual variability (Figure S1 in





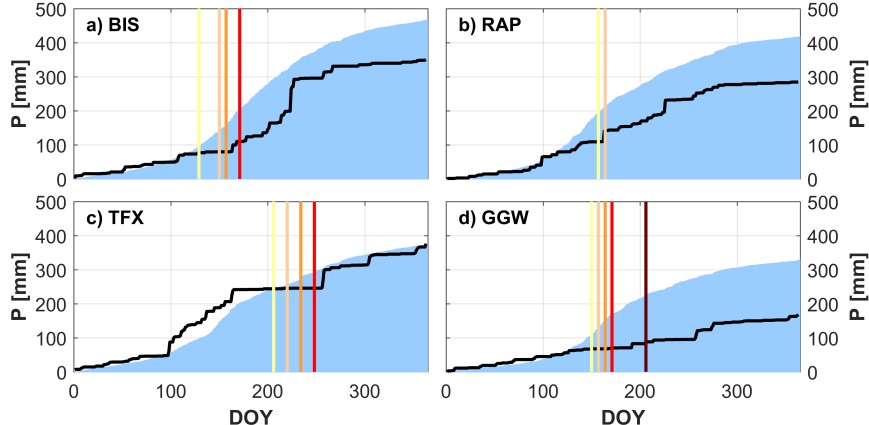

**Figure 2.** Cumulative annual precipitation for 2017 (black line) compared to 30-year average precipitation (1987-2016, shaded area) for (a) Bismarck, ND; (b) Rapid City, SD; (c) Great Falls, MT; and (d) Glasgow, MT. Vertical lines indicate onset of drought (D0 -– yellow to D4 — dark red).

Supplement). During the first 100 days of 2017, precipitation totals were close to the climatological mean at all four stations. After that, BIS, RAP, and GGW reported below-average precipitation between April and August. At GGW, there was almost no reported precipitation during spring and summer 2017. TFX, which initially reported above average precipitation in April and May, experienced a three-month period from late June to September (c. DOY 175-250) with virtually no precipitation
(Fig. 2).

In addition to $P$, $T$ (Figure S2), and the USDM (Fig. 1), drought development in the NGP can also be investigated from the atmospheric perspective. Applying the CTP-HI$_{low}$-framework (Findell and Eltahir, 2003a, b) reveals strong atmospheric control on precipitation in May (Fig. 3) preceding the USDM onset of *severe drought* (D2) conditions in the NGP. At GGW, near the centroid of the region that experienced *exceptional drought* (Fig. 1), average HI$_{low}$ for May 2017 was nearly 25°C.
This far exceeded the value of 20°C beyond which convective precipitation is unlikely. May 2017 conditions were more like a typical August, when rainfall is less likely (Fig. 2).

Across the four study sites, monthly mean CTP increased from less than $0\,\mathrm{J\,kg^{-1}}$ in April 2017 to the 150-250 $\mathrm{J\,kg^{-1}}$ range in September, indicating conditions during which convective development is not limited by atmospheric stability. At the same time, average monthly HI$_{low}$ values increased, indicating increasing atmospheric dryness, which can limit convective
precipitation (Findell and Eltahir, 2003a, b). Atmospheric profiles during July and August in the NGP are very dry on average and little convective precipitation is expected during an average year (Fig. 3), especially for TFX and GGW.

The dryness of atmospheric profiles (via HI$_{low}$) during 2017 preceded surface drought conditions in the NGP (Tab. 1, Figs. 1 + 3). At BIS, D0 conditions started on May 9, whereas the percentage of days characterized by convective suppression, indicated by HI$_{low} > 20$°C, exceeded the 75th percentile already in late March (Fig. 4). A similar situation arose for GGW,
which was most affected by the drought: the probability of convective precipitation was anomalously low before and during





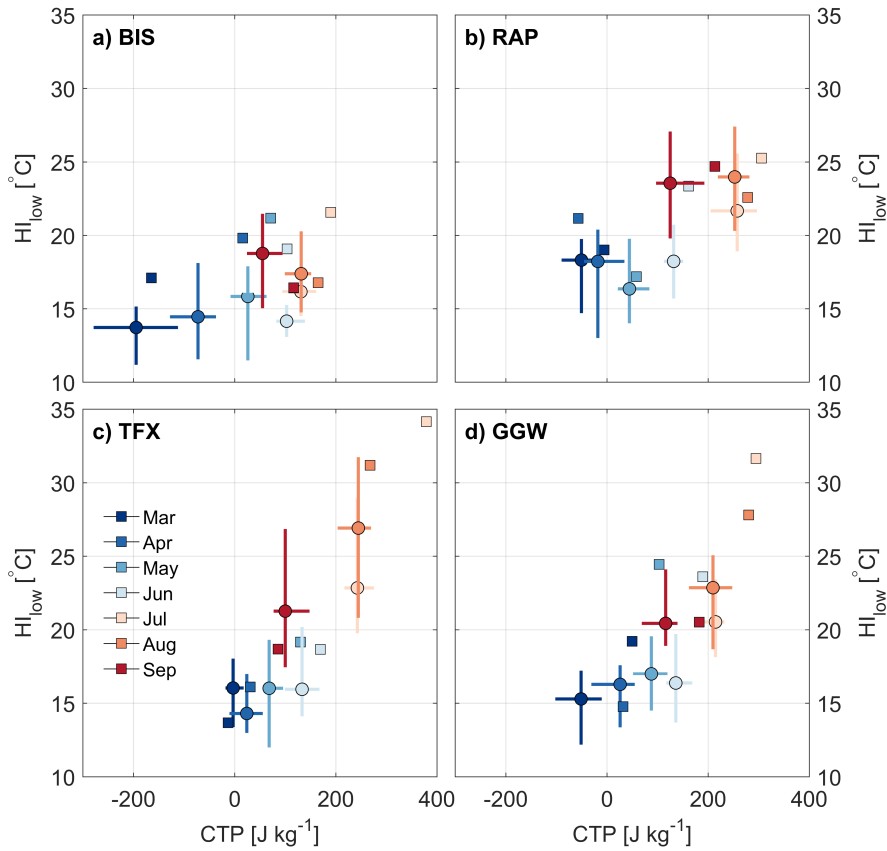

**Figure 3.** Seasonal development of mean monthly convective triggering potential (CTP) and lower tropospheric humidity index (HI$_{low}$) for 2017 (squares) compared to 30-year (1987-2016) median (circles) and interquartile range (lines) of CTP and HI$_{low}$ for (a) Bismarck, ND (BIS); (b) Rapid City, SD (RAP); (c) Great Falls, MT (TFX); and (d) Glasgow, MT (GGW).

May 2017 (Figs. 3 + 4d), yet the USDM showed May 30 and June 06 for D0 and D1 conditions, respectively. RAP, on the other hand shows the smallest lead in HI$_{low}$ compared to the other sites. May conditions were close the climatological mean, which was followed by rapid atmospheric drying during June and corresponded to the onset of D0 conditions on June 06. RAP did not experience D2 or greater conditions during 2017.

## 4   Discussion

Our findings suggest that a careful study of convective likelihood may provide further information for regional drought forecasts. Convective precipitation was far less likely than usual at BIS and GGW before the flash drought in 2017 as revealed by the monthly and daily CTP-HI$_{low}$ analyses (Figs. 3 + 4). The running mean of days in which convective precipitation was un-





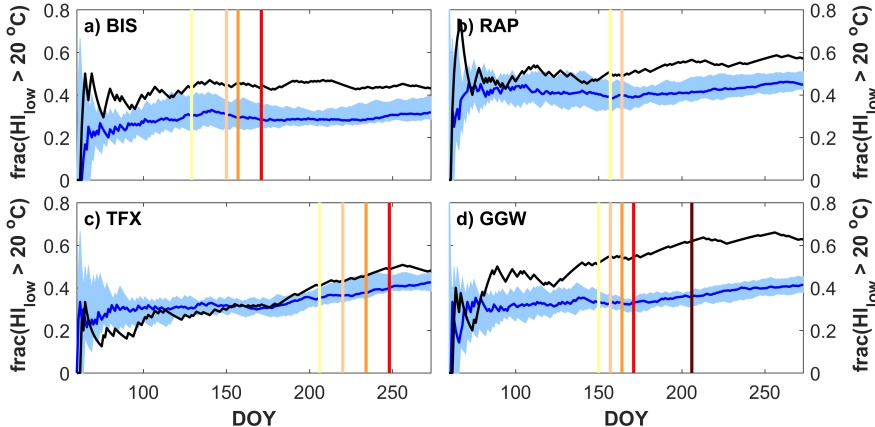

**Figure 4.** Cumulative fraction of days from March 1 (DOY = 60) to September 30 with the lower troposphere humidity index $HI_{low}$ exceeding 20°C - taken to be the value above which convective precipitation is unlikely - for 2017 (black line) compared to the 30-year median (blue line) and interquartile range (shaded area) for (a) Bismarck, ND; (b) Rapid City, SD; (c) Great Falls, MT; and (d) Glasgow, MT. Vertical lines indicate onset of drought (D0 — yellow to D4 — dark red).

likely (Fig. 4) suggests that GGW and BIS experienced anomalously dry conditions in mid-late March ("Anamolous" is defined here as the fraction of days during which the likelihood of convective precipitation exceeded the climatological interquartile range). In other words, there were atmospheric clues that conditions were drier than normal before the flash drought across part of the domain that experienced drought conditions (Fig. 1).

5      Observations that convective precipitation was anomalously low well before drought onset (Fig. 4) is consistent with the notion that flash droughts are preceded by a breakdown in surface-atmosphere feedbacks, as has been found in other studies of different drought events. For example, Myoung and Nielsen-Gammon (2010a, b) demonstrated the instrumental role of low soil moisture and associated convective inhibition on drought in Texas, and Juang et al. (2007a) noted a threshold in soil moisture and RH below which convective precipitation did not form in the Piedmont region of North Carolina. In the NGP,

10 convective precipitation rarely occurs at near-surface relative humidities below ca. 50-60% (Gerken et al., 2018), which may provide an approximate "rule of thumb" for convective precipitation likelihood in areas like the NGP where sounding stations are separated by hundreds of kilometers (Fig. 1). The present study likewise shows that there are clues in the coupled surface-atmosphere system that indicate when a flash drought may emerge. Such clues should be studied in concert across multiple known flash drought events to establish effective metrics to estimate flash drought onset.

15      It remains to be seen if the CTP-$HI_{low}$ analysis explored here is a robust method for forecasting flash droughts in other regions and for other drought events, and a critical course of future research is to do so in the context of other approaches for studying drought onset. For example, the evaporative stress index was anomalously high before the 2012 drought in the central Great Plains (Otkin et al., 2016) and VPD and $ET_p$ were anomalously high before flash droughts in the eastern U.S. (Ford and





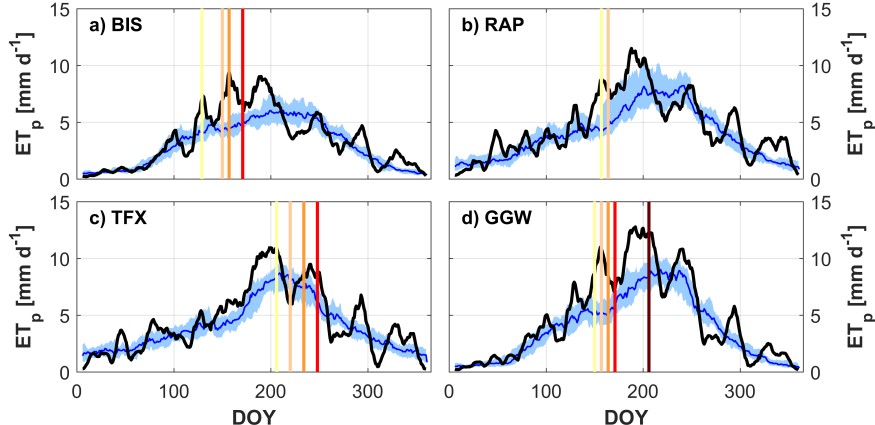

**Figure 5.** Estimated $ET_p$ for 2017 (black line), compared to 30-year median (blue line) and interquartile range (shaded area) for (a) Bismarck, ND; (b) Rapid City, SD; (c) Great Falls, MT; and (d) Glasgow, MT. Vertical lines indicate onset of drought (D0 — yellow to D4 — dark red). $ET_p$-values were smoothed by applying a 10-day running mean.

Labosier, 2017). Successive desiccation of soils, leading to tall, warm, and dry atmospheric convective boundary-layers was found to be an important contributor to heat waves (Miralles et al., 2014).

A similar connection between evaporative demand and drought is also found in during spring and summer in the NGP (Figs. 5 + S4). All four sites transitioned to D0 conditions during periods for which $ET_p$ exceeded the climatologically median

by at least one quartile. Drought intensification at BIS, RAP, and GGP occurred during an extended period of high evaporative demand, which could not be met by the typically dry soils. $ET_p$ was not anomalously high for extended periods before the onset of drought at BIS and RAP, but was greater than average for 25 days before the onset of D0 drought at GGW, and for an even longer period before D0 drought at TFX, which was declared after D4 exceptional drought conditions emerged at GGW (Tab. 1) and drought mitigation was underway in the State of Montana.

The unusually high 2017 $ET_p$ could be primarily attributed to it's sensitivity to VPD (Fig. 6). At the same time, wind speeds at GGW and BIS also play a role in elevated $ET_p$ values, whereas near-surface $T$ was found to be less important. Since no direct observations of historic $R_n$ were available at the stations, such that $R_n$ for $ET_p$ calculation was estimated primarily from solar position and temperature as suggested by FAO, it is not feasible to assess the role of abnormal energy input on $ET_p$. However, due to high VPDs and the development of general drought conditions, it can be presumed that cloud cover

was low, thus leading to larger than normal solar irradiance. This, likely, further increased $ET_p$, highlighting the importance of land-atmosphere feedbacks in drought development.

The interaction between drying soils, boundary-layer processes, evaporative demand, and convection for drought is also in agreement with findings that the timing of convection initiation in the Canadian Prairie Provinces is influenced by soil moisture (Hanesiak et al., 2004). Also, CPC soil moistures (Fan and van den Dool, 2004) became anomalously low (rank below 30th

percentile) in the Dakotas and northeastern Montana from June 2017 onwards. Given the difficulties of providing accurate





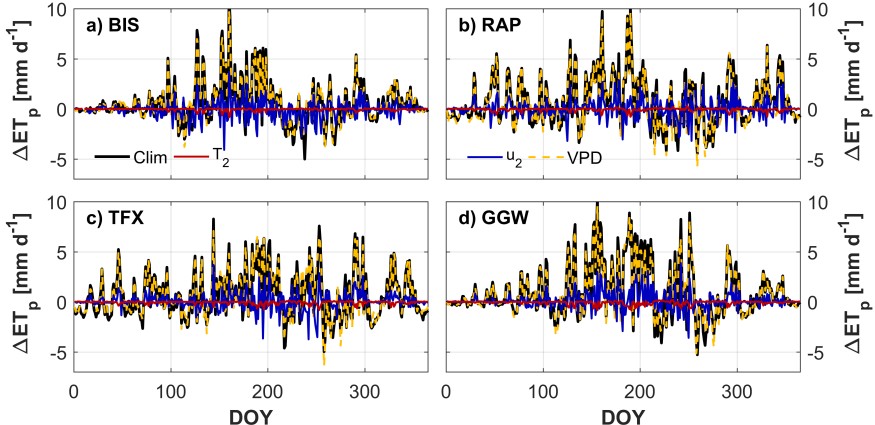

**Figure 6.** Change of $ET_p$ compared to estimated $ET_p$ of 2017 ($\Delta ET_p$) for (a) Bismarck, ND; (b) Rapid City, SD; (c) Great Falls, MT; and (d) Glasgow, MT. $\Delta ET_p$ is calculated by using the climatological averages of $T_2$, $u_2$, and VPD, respectively, instead of 2017 values, while *Clim* corresponds to $ET_p$ calculated from the climatology. $\Delta ET_p$ values were smoothed by applying a 10-day running mean.

root zone soil moistures, despite recent advances in observations and modeling, the CTP-HI$_{low}$ framework might provide an additional early warning signal for drought forecasting.

There are a number of additional factors at play that should be studied in concert with the likelihood of convective precipitation for a comprehensive understanding of the hydroclimatic conditions that precede flash drought in the NGP and elsewhere. Mesoscale convective systems provide a large fraction of total summertime precipitation in the Great Plains (Carbone and Tuttle, 2008; Tuttle and Davis, 2006), and the conditions that favor or suppress them provide an important control over drought development. Flash drought forecasting should also be cognizant of shifting land surface conditions that may have important consequences for the coupled soil-vegetation-atmosphere system (Otkin et al., 2016). Recent shifts in climate and precipitation in the NGP for example are consistent with shifts in agricultural management that impact soil moisture and boundary-layer dynamics (Gameda et al., 2007; Betts et al., 2013, 2014; Mahmood et al., 2014; Vick et al., 2016) such that the 30-year mean climatological conditions studied here are part of an ongoing trend in land-atmosphere coupling. Recent studies also suggest that the arid-humid divide across the North American Great Plains - often approximated as the 100th meridian in the U.S. - may be shifting eastward in response to global climatic changes (Seager et al., 2018). Regardless, atmospheric conditions favoring convective suppression preceded the 2017 NGP flash drought and may have provided robust inference into its prediction.

## 5 Conclusions

Our findings show that the NGP flash drought was exacerbated by a breakdown of land-atmosphere interactions and predominance of atmospheric control on convective initiation during the early growing season that countered the decadal trend of moistening in the NGP and increased land-atmosphere coupling (Gerken et al., 2018). Ruddell and Kumar (2009) similarly



found drought to be associated with a breakdown of the self-organizing moisture feedback process between surface and atmosphere. In the light of recent changes in land management for the NGP away from summer fallow and investment towards more water demanding crops (Miller et al., 2002; Long et al., 2014), reliable drought early warning systems become increasingly important. Integrating information about convective inhibition into drought monitoring and drought early warning systems

appears to be advisable based on this study, especially in regions characterized by high or variable degrees of land-atmosphere coupling where land surface and atmospheric observations provide information about precipitation likelihood. A critical course of future research is to compare the conditions that precede flash droughts using multiple metrics in order to provide the best possible inference for drought forecasts in the U.S. and across the globe.

*Competing interests.* The authors declare no competing interests.

*Acknowledgements.* We acknowledge support from the National Science Foundation (NSF) Office of Integrated Activities (OIA) 1632810, the NSF Division of Environmental Biology (DEB) 1552976, the U.S. Department of Agriculture National Institute of Food and Agriculture (USDA-NIFA) Hatch project 228396, the Montana Wheat and Barley Committee, and The Graduate School at Montana State University. BLR acknowledges support from NSF-Emerging Frontiers (EF) 1241960. Data to reproduce figures is available to the public from the National Oceanic and Atmospheric Administration's (NOAA) Global History Climatology Network (GHCN, ftp://ncdc.noaa.gov/pub/data/ghcn/daily/),

NOAA's Integrated Surface Dataset (ISD, https://www.ncei.noaa.gov/data/global-hourly/), and the University of Wyoming (http://weather.uwyo.edu/upperair/). We thank Larry Oolman for maintaining this service. The U.S. Drought Monitor (USDM) is jointly produced by the National Drought Mitigation Center at the University of Nebraska-Lincoln (NDMC-UNL), the United States Department of Agriculture, and the National Oceanic and Atmospheric Administration. Map information courtesy of NDMC-UNL. USDM data in the form of shape-files was obtained from: http://droughtmonitor.unl.edu. We thank Tanja Fransen (National Weather Service Forecast Office

Glasgow, MT), and Buddhi Achhami, Adam Cook, Hannah Goemann, Audrey Harvey, and Jim Junker (Montana State University) for helpful comments on a draft version of this manuscript.



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
