# Peer review of "Convective suppression before and during the United States Northern Great Plains Flash Drought of 2017"

_Hydrology and Earth System Sciences, 2018_

## Referee Comment (RC1) · Anonymous Referee #1 · 29 May 2018

General comments:

This is a nice paper that is a good fit for HESS. The authors demonstrate the importance of atmospheric humidity deficits just before the onset of the severe drought in the Northern Great Plains (NGP) of the United States during 2017. I do have a few suggestions that I believe will improve the analysis, and some minor comments to improve the flow and readability of the document. I believe these are minor suggestions, and I recommend publication after these issues are addressed.

1. Much of the interpretation of the figures is discussed in the text by referring to the month, but the figures use DOY on the x-axis. This forces the reader to do unnecessary

work. I recommend re-doing Figures 2, 4, 5, 6, S1, S2, S3, and S4 with the months clearly indicated on the x-axis.

2. Some of the analysis centers on the use of potential evaporation. Two papers by Milly and Dunne (Nature Climate Change 2016, Journal of the American Water Resources Association 2017) highlight the dangers and pitfalls of many empirical formulas for Ep. They show that the best method (particularly moving into the future) is the simplest: just 80% of net radiation. Given those results, I believe you should re-compute your Ep-based calculations. However, your discussion on page 8 about the impact of estimating Rn since there are no direct observations is still highly relevant, perhaps even more relevant. I do wonder how Figure 5 would change with better Ep estimates.

3. Figure 4 is a very nice figure! This figure really demonstrates how different the conditions at TFX are from the other stations. I would like to see more focus on the temporal relationship between the anomalous 2017 behavior shown in this plot and the period of peak climatological rainfall occurrence at each station, which can be deduced from Figure 2. BIS and GGW both show a lengthy run of atmospherically dry days (where the black line in Figure 4 exceeds climatological norms) prior to the onset of the rainiest period (the dramatic increase in the slope of the blue region in Figure 2). Thus, at these stations, drought onset begins and intensifies right away. At the other two stations, the climatologically rainiest time period gets underway prior to the intrusion of atmospherically controlled conditions indicated in June and July in Figure 3 and Figure 4, and the drought is not as severe (RAP) or begins much later (TFX). I think these differences between the stations should be brought forth more in this paper in order to more effectively make your point about the importance of convectively-unfavorable atmospheric conditions driving rapid drought onset.

Minor comments and edits:

1. Page 1, Line 21: "late May 2017": I think early June is more consistent with the Figure and Table.

2. Page 4, Line 3: You should explicitly mention the local time that is equivalent to 12 UTC.

3. Page 4, Line 31, "at all four soundings": I noted here that TFX is not drier than climatology until day 200. This is the first indication that behavior at TFX is really quite different than at the other stations.

4. Page 5, Line 8, "in May": at stations BIS and GGW (not all four).

5. Page 5, Line 19: include "climatological" before 75th percentile.

6. Figure 3: I had to look at this figure for a long time to sort out which months were anomalous at each station. I think this was simply because the many blue months were hard to distinguish from each other and because the July symbols sometimes got hidden under the August symbols and lines. Perhaps you can change your color choices to make it easier to quickly read these plots.

7. Page 8, line 1: "tall" should be "deep."

8. Figure S1: I didn't find this figure particularly helpful. It seems to me that the form is largely dictated by the nature of the denominator since Psum starts at zero and can only increase. I think the figure can be removed.

---

## Referee Comment (RC2) · Anonymous Referee #2 · 12 Jun 2018

General Comments: This work demonstrates the relevance of convective inhibition and convection suppression for rapid drought intensification in the NGP region. This work is particularly relevant given the heightened awareness of flash drought and the knowledge gap in our understanding of its drivers. Overall the analysis, writing, and presentation are of high quality. Therefore I recommend acceptance of this manuscript for publication after consideration of the following (minor) suggestions.

Specific Comments: 1) Because of the importance of high quality ETp estimates to the manuscript results, it would be nice to get an understanding of any error or bias in the Rn estimates (based on daily temperature range). The Bismarck airport ASOS

station does have solar radiation observations - as part of the National Solar Radiation Database (http://rredc.nrel.gov/solar/old_data/nsrdb/) - spanning 1961 to 2010. These direct observations can be used to get an idea for error or bias in your Rn estimates, and perhaps even some analysis of how these errors propagate when computing ETp.

2) Figure 5: I like the comparison of 2017 ETp to the climatology at each station; however, the daily ETp line is quite noisy, and it makes it difficult to see the 2017 absolute deviation from "normal". Could you perhaps show cumulative ETp over the course of the year instead? I think this would provide more insight as to how much larger evaporative demand was in 2017.

3) It is mentioned in multiple places throughout the manuscript that convective inhibition or convection suppression is important for flash drought monitoring or can be useful for drought early warning. This is supported by the results of this manuscript; however, these statements come with the significant caveat of a sample size of 1 (2 if you count the related Myoung and N-G studies). Any broader conclusions regarding the importance of convection suppression for rapidly intensifying drought cannot be made without more analysis of historical drought events. I don't think that this analysis is necessary for this manuscript, but any inference of the ability of convective inhibition (or HIlow) to improve flash drought forecasts based on this work should be made with this caveat in mind. I would like to see this limitation mentioned explicitly in either the discussion or conclusions section (preferably both).

Technical Comments: 1) Page 2, line 3: I think you should replace "consequences" here with "impacts"

2) Page 3, line 4: add "land-atmosphere" in front of coupling

3) Page 3, equation 1: although not as important here as HIlow, it would be nice to include the CTP equation as well.

4) Page 4, line 8: do you mean "Table 1" instead of "Tab 1"?

5) Page 4, line 10: I think you can delete "for data" here

6) Page 5, line 7: perhaps "boundary layer perspective" would be more precise than "atmospheric perspective"?

7) Page 8, line 1: replace "tall" with "deep"
* * *

---

## Author Comment (AC1) · 17 Jun 2018

**1 Response to Reviewer 1**

**1.1 General comments:**

This is a nice paper that is a good fit for HESS. The authors demonstrate the importance of atmospheric humidity deficits just before the onset of the severe drought in the Northern Great Plains (NGP) of the United States during 2017. I do have a few suggestions that I believe will improve the analysis, and some minor comments to improve

the flow and readability of the document. I believe these are minor suggestions, and I recommend publication after these issues are addressed.

We thank the reviewer for their work and the supportive comments. Please find below answers and changes to the text made in response.

1. Much of the interpretation of the figures is discussed in the text by referring to the month, but the figures use DOY on the x-axis. This forces the reader to do unnecessary work. I recommend re-doing Figures 2, 4, 5, 6, S1, S2, S3, and S4 with the months clearly indicated on the x-axis.

Thank you for pointing this out. We have updated the figures as recommended.

2. Some of the analysis centers on the use of potential evaporation. Two papers by Milly and Dunne (Nature Climate Change 2016, Journal of the American Water Resources Association 2017) highlight the dangers and pitfalls of many empirical formulas for Ep. They show that the best method (particularly moving into the future) is the simplest: just 80% of net radiation. Given those results, I believe you should re-compute your Ep-based calculations. However, your discussion on page 8 about the impact of estimating Rn since there are no direct observations is still highly relevant, perhaps even more relevant. I do wonder how Figure 5 would change with better Ep estimates.

Thank you for the paper suggestions. As both Reviewer 1 and Reviewer 2 pointed out $ET_p$ is highly dependent on the availability of energy (i.e. net radiation), which is commonly not available, especially over the long term. We do not have reliable net radiation measurements at all sites across the entire study period and estimating $ET_p$ as a fraction of net radiation would be highly sensitive to the net radiation estimate. For this reason we decided that it is best to keep using the standard method by the FAO to estimate $ET_p$ and added additional discussion to the paper: "The interpretation of $ET_p$ effects and sensitivity to drought development should be made cognizant of the fact that $R_n$ was estimated and thus subject to associated uncertainties. At the same time, recent work (Milly & Dunne, 2016; Milly & Dunne 2017) found Penman-Monteith

based ET$_p$ estimates to be overly sensitive in response to climate change. Based on the reported low cloud cover during the drought, it is likely that the impact of energy supply is underrepresented in the current work." Thank you for pointing out the recent papers by Milly & Dunne.

3. Figure 4 is a very nice figure! This figure really demonstrates how different the conditions at TFX are from the other stations. I would like to see more focus on the temporal relationship between the anomalous 2017 behavior shown in this plot and the period of peak climatological rainfall occurrence at each station, which can be deduced from Figure 2. BIS and GGW both show a lengthy run of atmospherically dry days (where the black line in Figure 4 exceeds climatological norms) prior to the onset of the rainiest period (the dramatic increase in the slope of the blue region in Figure 2). Thus, at these stations, drought onset begins and intensifies right away. At the other two stations, the climatologically rainiest time period gets underway prior to the intrusion of atmospherically controlled conditions indicated in June and July in Figure 3 and Figure 4, and the drought is not as severe (RAP) or begins much later (TFX). I think these differences between the stations should be brought forth more in this paper in order to more effectively make your point about the importance of convectively-unfavorable atmospheric conditions driving rapid drought onset.

We would like to thank the reviewer for their supportive comment. We agree with his suggestion that there are important differences between BIS, GGW, TFX, and RAP with respect to rapid drought intensification. Both BIS and GGW are part of the region with an early and strong drought onset, whereas TFX is only affected later and RAP is at the edge of the severe drought region. We have added additional discussion with respect to Figures 2 and 4 to further highlight this in the text. The first paragraph of the discussion contains: "Notably, this period of low convective likelihood coincided with the period of the vegetative growing season which is climatologically wettest (Fig.2). At the other stations in contrast, such conditions either occur much later (TFX) or are less severe (RAP) thus limiting rapid drought intensification and severity. This differing

behavior further highlights the likely importance of convectively unfavorable conditions and atmospheric control for drought."

1.2   Minor comments and edits:

1.  Page 1, Line 21: "late May 2017": I think early June is more consistent with the Figure and Table.

The drought was first indicated in the May 30 USDM. We changed the text to "late May to early June".

2. Page 4, Line 3: You should explicitly mention the local time that is equivalent to 12 UTC.

Thank you. We added: "(corresponding to 5:00 MST)"

3.  Page 4, Line 31, "at all four soundings": I noted here that TFX is not drier than climatology until day 200. This is the first indication that behavior at TFX is really quite different than at the other stations.

Thank you for pointing this out. Please see our response to *General Comment 4* for further detail and changes to text.

4. Page 5, Line 8, "in May": at stations BIS and GGW (not all four).

Thank you for this comment. We added an explicit reference to BIS and GGW to the sentence.

5. Page 5, Line 19: include "climatological" before 75th percentile.

Added, thank you.

6. Figure 3: I had to look at this figure for a long time to sort out which months were anomalous at each station.  I think this was simply because the many blue months

were hard to distinguish from each other and because the July symbols sometimes got hidden under the August symbols and lines. Perhaps you can change your color choices to make it easier to quickly read these plots.

Thank you for the comment. During preparation of the figure, we tried several different color schemes and generally found that finding 7 different colors that are easily distinguishable, while also avoiding red and green in the same map, is very difficult. Also we feel that using blue for March and reddish for August and September, gives a good contrast.

7. Page 8, line 1: "tall" should be "deep."

Thank you for pointing this out, we changed the passage.

8. Figure S1: I didn't find this figure particularly helpful. It seems to me that the form is largely dictated by the nature of the denominator since Psum starts at zero and can only increase. I think the figure can be removed

The reviewer is correct that the shape of the curves is determined by an increasing denominator. This is why the figure was placed in the supplement. The intention of Figure S1 is to give some estimate of climatological interannual variation for Figure 1 and we would like to retain the figure there for reference.

---

## Author Comment (AC2) · 17 Jun 2018

**1  Response to reviewer 2**

**1.1  General Comments:**

This work demonstrates the relevance of convective inhibition and convection suppression for rapid drought intensification in the NGP region. This work is particularly relevant given the heightened awareness of flash drought and the knowledge gap in our understanding of its drivers. Overall the analysis, writing, and presentation are of

high quality. Therefore I recommend acceptance of this manuscript for publication after consideration of the following (minor) suggestions.

We thank the reviewer for their work and the supportive comments. Please find below answers and changes to the text made in response to their thoughtful suggestions.

1.2   Specific Comments:

1) Because of the importance of high quality ETp estimates to the manuscript results, it would be nice to get an understanding of any error or bias in the Rn estimates (based on daily temperature range). The Bismarck airport ASOS station does have solar radiation observations - as part of the National Solar Radiation Database (http://rredc.nrel.gov/solar/old_data/nsrdb/) - spanning 1961 to 2010. These direct observations can be used to get an idea for error or bias in your Rn estimates, and perhaps even some analysis of how these errors propagate when computing ETp.

Thank you for pointing out this interesting dataset. Because our research in flash droughts and the NGP drought is ongoing, we are thankful for any additional data sources. The analysis demonstrated that $ET_p$ from the well-established FAO framework was not anomalously high or low before the NGP flash drought and would not have provided skill in forecasting the drought. This does not mean that it may not provide inference for future flash droughts, and we will consider this database for future studies.

2) Figure 5: I like the comparison of 2017 ETp to the climatology at each station; however, the daily ETp line is quite noisy, and it makes it difficult to see the 2017 absolute deviation from "normal". Could you perhaps show cumulative ETp over the course of the year instead? I think this would provide more insight as to how much larger evaporative demand was in 2017.

During preparation of the manuscript we discussed whether to include $ET_p$ sums or daily values and decided on the former, since daily $ET_p$ is small compared to the total
annual sum. Consequently, changes on monthly scales are not readily seen. We have already included the annual sum figure in the supplement to this manuscript (Fig. S4). We have now made the reference to the supplement more explicit in the text.

3) It is mentioned in multiple places throughout the manuscript that convective inhibition or convection suppression is important for flash drought monitoring or can be useful for drought early warning. This is supported by the results of this manuscript;however, these statements come with the significant caveat of a sample size of 1 (2 if you count the related Myoung and N-G studies). Any broader conclusions regarding the importance of convection suppression for rapidly intensifying drought cannot be made without more analysis of historical drought events. I don't think that this analysis is necessary for this manuscript, but any inference of the ability of convective inhibition (or HIlow) to improve flash drought forecasts based on this work should be made with this caveat in mind. I would like to see this limitation mentioned explicitly in either the discussion or conclusions section (preferably both).

Thank you for noting this, we were careful to avoid general statements regarding all droughts in the text. In response to this comment, we added to the discussion: "However, our findings are derived from a single flash drought event. To generalize our findings and to potentially establish effective metrics to estimate flash drought onset, the historic record of flash droughts should be investigated to detect similar clues."
For the conclusion we modified the sentence in question to: "Integrating information about convective inhibition into drought monitoring and drought early warning systems might be advisable based on this study. A critical course of future research and to generalize our findings beyond a single case is to compare the conditions that precede flash droughts using multiple metrics in order to provide the best possible inference for drought forecasts in the U.S. and across the globe.

1.3 Technical Comments:

1) Page 2, line 3: I think you should replace "consequences" here with "impacts"

Thank you for pointing this out, we took the recommendation.

2) Page 3, line 4: add "land-atmosphere" in front of coupling

Added, thank you.

3) Page 3, equation 1: although not as important here as HIlow, it would be nice to include the CTP equation as well.

Thank you for the comment. Interestingly, we are not aware of a paper that has included the equation for CTP. The equation and explanation were added: '

$$\mathrm{CTP} = \int_{100\,\mathrm{hPa}}^{300\,\mathrm{hPa}} R_d \; (T_{v,sp} - T_{v,e}) \; \mathsf{d}\ln p \qquad (1)$$

with $T_{v,sp}$ and $T_{v,e}$ as virtual temperatures of a saturated air parcel and the environment, the specific gas constant for dry air ($R_d$, and pressure ($p$).'

4) Page 4, line 8: do you mean "Table 1" instead of "Tab 1"?

Thank you. To avoid confusion, we now use "Fig." and "Tab." as abbreviations throughout the text.

5) Page 4, line 10: I think you can delete "for data" here

Deleted, thank you for the suggestion.

6) Page 5, line 7: perhaps "boundary layer perspective" would be more precise than "atmospheric perspective"?

We added 'boundary-layer' to the text, leaving 'atmospheric' in as well since strictly speaking convection goes beyond the ABL.

7) Page 8, line 1: replace "tall" with "deep"

We changed the text, thank you for the suggestion.

---

## Author Comment (AC3) · 18 Jun 2018

For convenience we uploaded a PDF (see supplement to this comment) with the proposed modifications highlighted in the text (blue).

Please also note the supplement to this comment: https://www.hydrol-earth-syst-sci-discuss.net/hess-2018-211/hess-2018-211-AC3-supplement.pdf